# Study on the Shear Behaviour and Fracture Characteristic of Graphene Kirigami Membranes via Molecular Dynamics Simulation

**DOI:** 10.3390/membranes12090886

**Published:** 2022-09-14

**Authors:** Yuan Gao, Shuaijie Lu, Weiqiang Chen, Ziyu Zhang, Chen Gong

**Affiliations:** 1School of Transportation and Civil Engineering, Nantong University, Nantong 226019, China; 2Department of Mechanical, Aerospace and Civil Engineering, School of Engineering, The University of Manchester, Manchester M13 9PL, UK; 3Department of Chemical Engineering and Analytical Sciences, The University of Manchester, Manchester M13 9PL, UK

**Keywords:** membrane science, graphene kirigami, MD simulation, shear performance, porosity

## Abstract

In this study, we aimed to provide systematic and critical research to investigate the shear performance and reveal the corresponding structural response and fracture characteristics of the monolayer GK membrane. The results demonstrate that the kirigami structure significant alters the shear performance of graphene-based sheets. Tuning the porosity by controlling the incision size, pore distribution, and incision direction can effectively adjust the shear strength and elastic modulus of GK membranes. The trade-off of the stress and strain of the GK membrane is critical to its shear behaviour. The microstructural damage processes and failure characteristics further reveal that making more carbon atoms on the GK structure sharing the strain energy is the key to reinforcing the shear performance of membranes. Based on this, we found that adding the shear loading in the direction of perpendicular to the incisions on the GK membrane can significantly improve the shear strength and stiffness of the membrane by 26.2–32.1% and 50.2–75.3% compared to applying shear force parallel to GK incisions. This research not only broadens the understanding of shear properties of monolayer GO membrane but also provides more reference on the fracture characteristics of GK membranes for future manufacturing and applications.

## 1. Introduction

By virtue of its monolayer honeycomb dense structure [1], excellent chemical stability [2], superior electrical and mechanical properties [3], graphene-based membranes have been widely investigated in the last decade [4]. Recently, graphene and graphene oxide membranes with micro- or nano-pores have attracted extensive research attention of researchers, which own great potential application prospects in a broad spectrum of fields, including gas separation and purification [5,6], CO_2_ capture and storage [7,8], water desalination and treatment [9,10], DNA sequencing [11,12], hazardous waste containment system [13], chemical processing, bionics [14,15], as well as pharmaceutical and medical applications [16,17]. However, the fixed pore size, weak bending stiffness and low stretchability always limit the applications of the nanoporous graphene (NPG) membranes [18,19].

To overcome these challenges, in the author’s previous study [20], we proposed using graphene kirigami (GK) as an ultra-membrane to endow graphene-based membranes with high adjustability. The concept of kirigami was inspired by ancient arts and derived from paper for cutting (kiru), folding (oru), and paper (kami) [21,22]. This technique transforms thin 2D materials into 3D materials to modify material and structure properties [23,24]. The latest theoretical and experimental study has verified that compared with the NPG membranes, GK membranes have ultra-high bending stiffness [25] and adjustability without damaging their structure [26,27]. Besides, GK membranes preserve the benefits of the NPG membranes with micro- or nano-scale defects, single-atom layer thickness, stable chemical properties and superior mechanical behaviour [20,28]. Moreover, due to the adjustable superiority, GK structures do not have rigorous pore geometry restrictions used as membranes, providing a great opportunity for future fabrication and application [28].

Understanding the mechanical properties of GK membranes is the premise of their application. To date, several studies have reported the mechanical performance of GK membranes using theoretical methods [25,29]. Qi et al. [29] utilized molecular dynamics simulation to perform that the yield and fracture strain of GK structures could be approximately a factor of three larger than that of standard monolayer graphene sheets. They verified that benefiting from the kirigami pattern, the GK structures could be held in the thinnest possible nanostructures. Bahamon et al. [30] further proposed that GK membranes could act as the platform for adjustable and stretchable quantum dot arrays. Gamil et al. [31] studied the shear flexibility of GK under shear loading via the theoretical analysis, revealing that the kirigami structure was highly efficient in tuning the shear behaviour of the monolayer graphene. Through the trade-off between strength and flexibility, the kirigami technique manipulated the shear performance of graphene-based materials.

Although several previous theoretical researches have explored the mechanical properties of the GK structure, few systematic studies have been performed on the shear behaviour and fracture characteristics of GK membranes. Furthermore, the exceeding potential of the GK membranes for a wide range of applications over other artificial and natural membranes was highlighted. Nevertheless, the influence of the incision size distribution, incision directions and surface pore densities on the shear behaviour and fracture characteristics of GK membranes have not been reported yet. Hence, in this study, we aimed to provide systematic and critical research to investigate the shear performance and reveal the corresponding structural response and fracture characteristics of the monolayer GK membrane. Firstly, the single-atom incised GK membranes were modelled according to the previous literature [20]. After that, the shear loading was simulated using LAMMPS software [32]. The effects of incision size, pore distribution, porosity, incision direction and loading gradient on the shear strength, strain and elastic modulus of the GK membranes were illustrated. The microstructural damage processes and failure characteristics of monolayer single-atom incised GK membranes were further discussed.

## 2. Methods

### 2.1. Model Construction

The configuration of the hydrogenated GK membranes with either zigzag or armchair incisions under shear loading was illustrated in Figure 1. In this study, we only considered the monolayer single-atom incised GK membranes. By virtue of the single-atom thickness, high stretchability and high adjustability, the monolayer GK membrane owns many properties that other single-layer graphene-based membranes do not have [20]. For example, the pore size of the NPG membrane is fixed. Therefore, in gas purification and water desalination applications, NPG membranes often need to adjust the pore size by overlapping multiple membranes [20,28]. By contrast, for monolayer GK membrane, the pore size could be controlled via adjusting the applied strains. Multi-layer arrangement of GK membranes is not necessary in most cases. For model construction processes, two-column of carbon atoms in the dotted boxes on the graphene sheets were excavated to form incisions, as shown in Figure 1a. After that, the hydrogen atoms were added to the deboning regions, and then the smallest GK unit was generated. Then, the GK unit was supercell to the lattice size of 51.12 × 49.2 Å^2^ with a periodic boundary. The geometries of the supercell GK membrane with zigzag incisions were demonstrated in Figure 1b. The porosity of the GK membranes was adjusted by the following two parameters: the incision length (L) and the distance between incisions (D), as shown in Figure 1b. The measured L and D of different zigzag-incised and armchair-incised GK membranes are listed in Table 1. The calculated porosity of the GK membrane was defined as the ratio of the excavated carbon atoms to the total atoms on the pristine graphene sheet, according to the previous report [32]. Here, we selected the porosity of the GK membranes between 6% and 12%, mainly based on their future application. Previous reports [20] indicated that the pore size of single-atom incised GK membranes was at the same level as some atoms and molecules, which could promote the practical application of GK membranes, such as gas purification and water desalination.

### 2.2. MD Simulation Details

Depending on the direction of the shearing deformation, i.e., being parallel or perpendicular to the incised direction, a thin layer of carbon atoms at the upper and lower edges (or left and right edges) were selected while the other region of the GK membranes was kept fully flexible. And then the left/lower edge was fixed while the right/upper was moved parallelly at a constant velocity (5 m/s) to impose the shearing force. This velocity is also used by the previous study [32] and has been proved sufficiently low to obtain the equilibrium results. Periodic boundary condition was applied in the direction normal to the shearing force. A NVT ensemble using a Nose-Hoover thermostat and remaining at a temperature of 300 K was adopted for our modelling with the timestep of 1 fs [33]. Even though a certain degree of residual stress will exist in the GK membranes after the relaxation stage using the NVT ensemble instead of the *NPT* ensemble (e.g., Figure 2), the usage of the NVT ensemble with membranes’ edges fixed is more closer to the practical nano-fabrication technology of the GK membranes (e.g., selective tearing [20]) than that of the *NPT* ensemble. To achieve an initial equilibrium state, the system firstly experienced a relaxation running of 1.0 ns with the upper and lower edges (or left and right edges) fixed, and then a 0.5–1.0 ns simulation run was conducted to dynamically apply the shearing loading. The simulation time of 1 ns for the relaxation stage is found to be sufficient to obtain the equilibrium results according to the convergence of the system’s potential energy within 1.0 ns. Our MD modellings adopted the reactive empirical bond order (REBO) potential [34], which had been found to accurately predict the mechanical responses of the graphene, e.g., [35,36]. Note that the improper selection of the interaction cut-off radius in REBO potential can cause non-physical strain hardening in stress-strain curves of carbon nanostructures. Therefore in our study, we have followed the suggestions from the previous studies [37,38,39] and set the cut-off radius to be 0.2 nm to effectively eliminate this non-physical strain hardening. The forces acting on the edge atoms were summed to calculate the shearing force. We repeated the simulation procedure for each case for three times with different initial velocities following a Gaussian distribution, to ensure the reproductivity. No significant deviations were observed among duplicated ones, and therefore only one typical data from three duplications was presented in our subsequent analysis. The representative LAMMPS input files can be found in the Appendix A.

## 3. Results and Discussion

### 3.1. Porosity Effect

The influence of the porosity on the shear behaviours of the GK membrane is significant. The porosity of the single-atom incised GK membrane can be tuned through two main approaches: adjusting the incision length and the spacing between incisions. Hence, in this section, the effects of the porosity considering the incision length (Group_1) and the spacing between incisions (Group_2) on the stress, elastic modulus, structural response, and fracture characteristics of the monolayer GK membranes under shear loading are studied. The stress-strain curves of zigzag-incised GK membranes in Group_1 are presented in Figure 2a, where four curves demonstrate a similar change trend. Under shear loading, the curve first entered the elastic stage, the stress and strain increasing linearly. When the strain reached 20%, the elastic modulus decreased, and the curve bent downward due to the deformation of the GK membranes. Continuing to increase the load, the strength of the GK membranes hit the peak value. After that, the GK structure failed, and the stress dropped rapidly. Nevertheless, the GK structure was not completely destroyed and could still bear a specific load, though the bearing capacity was far less than that of the structurally intact GK membranes. With the continued increase in strain, the GK membranes eventually disconnected completely. The calculated stress and elastic modulus of the zigzag-incised GK membranes are listed in Table 2 and Figure 2b. It can be found that with the increment of the porosity via rising the incision length, the stress and elastic modulus of the GK membranes exhibited a linear downward trend. The fitting coefficient was relatively high, ranging from 0.939 to 0.994.

The microstructural damage processes and failure characteristics of monolayer zigzag-incised GK membranes in Group_1 are demonstrated in Figure 3. For Sample ZZ_1_1, the deformation of the GK membrane was relatively uniform in the linear elastic stage, as shown in Figure 3a. The strain energy between carbon atoms was low except for the two fixed edges. With the continuous increase of the shear force and the growth to the ultimate failure stage (Stage II), apparent stress concentration appeared on the zigzag-incised GK membrane. Due to the narrow graphene unit between incisions, the stress concentration was mainly concentrated in these regions, resulting in considerable strain energy between carbon atoms, particularly at the edge of the incisions. The regions with high strain levels formed folds along the zigzag-incised GK membrane, roughly at a 45° to the shear edge and generating wrinkles. By contrast, the carbon atoms not on the winkles had significantly lower strain energy than that on the wrinkles. After the peak shear stress was reached, the zigzag-incised GK membranes would rupture at the incisions closest to the shear edge. Under this stage (Stage III in Figure 3a), except for the carbon atoms on the fracture surface, the strain energy of other carbon atoms on the film plummeted, and the GK membrane suffered severe uneven deformation. When the shear force was continuously increased, the GK membrane could still bear a specific load, and this part of the load was mainly borne by the graphene elements at the edge of the shear edge until the structure was broken entirely.

With the decrease of the porosity via reducing the incision length, the shear stress and elastic modulus of the zigzag-incised GK membranes showed an upward trend, rising from 20.7 GPa and 51.9 GPa under 11.6% porosity to 50.2 GPa and 126.3 GPa under 6.1% porosity. The enhancement of shear resistance was mainly due to the decrease in the incision length. More carbon atoms bore the strain energy during the shear loading, as presented in Figure 3. As the incision length was reduced to a certain extent, the destruction mode of the zigzag-incised GK membrane also changed. As the interval between incisions increased, the strain energy that could be borne was also enhanced, and it was not easy to be broken in parallel, as demonstrated in Stage IV in Figure 3a. Instead, the failure mode was a through-hole formed between two adjacent incisions on the edge of the sheared edge. In this failure mode, the stress drop of the zigzag-incised GK membrane was relatively low, and the whole membrane still owns a specific bearing capacity, as exhibited in Stage III of Figure 3d. The carbon atoms that initially suffered from high strain energy maintained a similar level, and the carbon atoms at the edge produced a more significant stress concentration until the membrane was utterly destroyed.

Another way to tune the porosity of the single-atom incised GK membrane was by changing the interval of the incision. To further investigate the influence of the incision interval length on the shear behaviour of the zigzag-incised GK membrane, Samples Group_2, as well as Sample ZZ_1_2, were considered. The stress-strain curves, stress strength and elastic modulus of zigzag-incised GK membranes in Group_2 are shown in Figure 4. Similar to the GK membranes in Group_1, with the increase of porosity, the shear stress of the specimen showed a decreasing trend, dropping from 33.7 GPa (10.0% porosity) to 22.4 GPa (19.9% porosity). Figure 4b further presented that the porosity of the zigzag-incised GK membranes in Group_2 had a quadratic function correlation with their peak strength and elastic modulus (*R*^2^ = 0.897 − 0.999). The difference was that the strain showed an opposite changing trend. When the porosity was 10%, the strain of the zigzag-incised GK membrane at the peak stress was only 0.42. However, when the porosity increased by 19.9%, the strain of the GK membrane at the peak stress grew to 0.64, rising approximately 52.4%. This simulation result indicated that more incisions in the zigzag-incised GK could efficiently reinforce the adjustability of the membranes.

Adjusting the porosity of the single-atom incised GK membrane by changing the incision interval also affected the structure’s failure mode, as presented in Figure 5. Sample ZZ_2_1 exhibited an utterly different failure mode from the samples in Group_1. The difference mainly came from two aspects. On the one hand, after Sample ZZ_2_1 reached its peak strength, the structure only suffered small-scale damage. Therefore, the partly damaged zigzag-incised GK membrane could still bear a particular load. As the load continued to increase, part of the GK membrane’s structure was again damaged. Experiencing three stress dropping, Sample ZZ_2_1 finally lost the bearing capacity. On the other hand, different from samples in Group_1, when the sample was finally destroyed, severe damage occurred at the edges of the two fixed edges, proving that in Sample ZZ_2_1, many carbon atoms were in the limit state of strain energy when they were destroyed. As the interval between incisions increased and the porosity reduced, the failure modes of Samples ZZ_2_2 and ZZ_2_3 were similar to those of the samples in Group_1, mainly destroyed at the shear edge.

### 3.2. Incision Direction Effect

Due to the unique chirality of graphene, the influence of incision direction on the mechanical properties of the GK membranes needs to be further explored. Previous studies reported that graphene had distinct electrical and thermal conductivity in different chiral directions [40,41]. Hence, this section revealed the effects of the incision direction on the shear behaviour. In order to better compare the shear behaviour and fracture characteristic of the GK membranes with the incision direction effect, we selected the armchair-incised and zigzag-incised GK structures with similar porosity for analysis. The stress-strain curves of the GK membranes with approximately 8.1–8.6% porosity (Samples AC_3_1, AC_3_2 and ZZ_1_3) and 9.8–11% porosity (Samples AC_3_3, AC_3_4 and ZZ_1_2) are presented in Figure 6. It could be found that with similar porosity, zigzag-incised GK membranes could withstand slightly higher shear strength than armchair-incised ones, while the armchair-direction incisions would assist in increasing the strain of the membranes.

Figure 7 further illustrates the structural stress and strain change factors by analyzing the failure modes of armchair-incised membranes. The failure mode of Sample AC_3_1 was similar to that of the zigzag-incised based group, both of which were due to the shearing action to form wrinkles on the membranes, contributing to the high strain energy of the carbon atoms on the wrinkles, which eventually led to destruction. With increasing porosity, armchair-incised GK membrane’s damage processes and failure characteristics radically changed. Figure 7b demonstrates the damage processes of Sample AC_3_2. When the specimen was subjected to peak shear strength, the regions between incisions formed wrinkles along the diagonal of the membrane, approximately 45° from the direction of the shear load. The carbon atoms in this region bore high strain energy, reaching 0.6–0.8 eV, while the carbon atoms in other regions sustained smaller strain energy. Subsequently, the specimen fractured at the connection between the incisions, and the stress produced a significant drop from 30.8 GPa to about 10.1 GPa, accompanied by a vast membrane deformation. The strain energy on the membrane was redistributed after rupture, and the whole membrane still had a specific shear resistance. After that, the shear strain was continuously applied to the Sample AC_3_2, and the whole specimen was continuously pulled along the failure defect until it was destroyed.

As the porosity continued to rise, the failure characteristics of the armchair-incised membranes produced a more considerable change. For Sample AC_3_3, there were two apparent peaks in the sample under the action of shear force, and the peak strength after the damaged structural part was higher, as presented in Figure 7c. Due to the extreme value of the strain energy of some carbon atoms at the fixed edge, Sample AC_3_3 suffered a small-scale failure at Stage I-II, and the stress produced an inevitable drop. However, since the main structure of the GK was integrated, the overall strain field of the carbon atoms on the membrane changed a little. As the shear loading to Stage III continued to increase, severe stress concentration occurred on the two fixed edges, some C-C bonds were broken, and the overall structure was destroyed. For Sample AC_3_4, different from other samples, the GK membrane entered a wave stage after the shear elastic stage. This was mainly due to breaking individual C-C bonds in Sample AC_3_4 under shearing action, resulting in some minor stress dropping and redistribution of strain energy. When the shear stress was increased to 21 GPa, a large-scale fracture occurred at the intermediate incision interval, and the structure was severely damaged. It should be noted that the armchair-incised GK membrane owned higher strain, up to 0.88–0.93 for Samples AC_3_3 and AC_3_4, 109–114% larger than that of zigzag-incised GK membrane under similar porosity.

### 3.3. Shear Loading Directions Effect

The direction of shear loading applied to the GK structure is another significant factor affecting the shear behaviour of the membranes. In Section 3.1 and Section 3.2, the shear force applied on GK membranes was always parallel to the incision direction. Therefore, this section applied the shear force along the direction perpendicular to the incisions of the GK membranes. For better comparison, we selected Sample ZZ_1_3 and Sample AC_3_2 to ensure other parameters were unchanged. The cases after changing the load direction were marked as Sample ZZ_1_3_P and Sample AC_3_2_P, as presented in Figure 8. It could be found that the shear performance of the GK membranes was significantly improved after the shear load was changed to be perpendicular to the incision direction. The shear stress increased from 32.5–36.5 GPa to 43–46 GPa, rising approximately 26.2–32.1%. The toughness of the GK membranes was also enhanced, with an increment of 50.2–75.3% from 51.1–94 GPa to 89.6–141.2 GPa. The improvement in shear properties had a more significant effect on the zigzag-incised GK membrane.

Figure 9 further exhibits the microstructural damage processes, and failure characteristics of the GK membrane loaded perpendicular to the direction of the incisions. Compared with Group_1 to Group_3, loaded perpendicular to the incisions of GK membranes, the sample exhibited a more complex post-peak variation. This was mainly because the failure of GK membranes was no longer along the shear edges but along the direction perpendicular to the wrinkles formed on the structure. Therefore, the rupture of the C-C bonds on the GK membranes progressed step by step, and the fluctuation after the peak was also more intense. For Sample ZZ_1_3_P, the rupture of the membrane was mainly divided into the following steps. With the growth of shear force, the strain energy was mainly concentrated on the formed wrinkles. The load-bearing strain of some C-C bonds reached the limit and rupture occurred, and a load of GK membranes also appeared to drop to a certain extent. Afterwards, a small-scale redistribution was performed on the strain energy on GK membranes. Due to the fracture of some C-C bonds, the strain that the GK membrane could withstand was also higher, and the stress was increased correspondingly. As the strain continued to increase, an obvious shear crack appeared in Sample ZZ_1_3_P (Stage VI-VIII in Figure 9a). With the expansion of the main crack, the strain of carbon atoms on the crack reached the limit and then broke, and the GK membrane was destroyed. Sample AC_3_2_P also exhibited a similar destruction mode as Sample ZZ_1_3_P. In this failure mode, the peak shear stress was higher than the carbon atoms subjected to the high strain, so the GK membranes had a higher shear strength and toughness than Group 1–3.

## 4. Conclusions and Outlooks

In this work, the shear behaviour and fracture characteristics of the single-atom incised GK membranes in both zigzag and armchair directions were studied via MD simulations. Our results indicated that when the porosity was varied by controlling the incision length, the shear strength and elastic modulus of GK membranes decreased linearly with increasing porosity. However, as the porosity on GK membranes was adjusted by changing the interval between incisions, although the shear strength and elastic modulus of GK membranes still dropped with the increase of porosity, they presented a quadratic function relationship. The failure mode of the zigzag-incised GK membrane was mainly shear-edge shear failure. For zigzag-incised GK membranes, the smaller interval between incisions was, the lower the shear strength of the membrane would be, but shear strain would grow. Under the same porosity, changing the incisions of the GK structure to the armchair direction would lead to a slight decrease in the shear strength of the membranes, weakening approximately 2.9–23.7%. Nevertheless, changing the direction of the shear loading on GK membranes could significantly improve the shear strength and stiffness of the membrane. When shear loading was applied perpendicular to the incisions on the GK membrane, the structure’s shear strength and elastic modulus increased about 26.2–32.1% and 50.2–75.3%, respectively. The improvement of shear performance was mainly due to the change of strain field and failure mode of the GK membranes. As the shear loading was applied perpendicular to the incisions on the GK membrane, more carbon atoms were subjected to high strain energy. After exceeding the limit strain energy, the GK membrane was gradually broken along the direction perpendicular to the wrinkles, so it also exhibited well post-peak bearing capacity.

The vital issue investigated in this work is the effects of the incision size, pore distribution, incision direction and shear loading direction on the shear behaviour of the GK membrane. Optimizing the shear performance of the GK membrane is still a challenge in materials design since many factors influence the shear behaviour. In future research, we decide to optimize the design of the GK membrane by adjusting various parameters to achieve better shear performance based on existing work and with the assistance of the machine learning method [42]. A machine learning method is a promising approach to screening over an ample design space, which has been successfully applied in designing solid electrolytes in lithium batteries [43] and predicting the thermal conductivity for nanomaterials [44]. Hence, we consider that the effects of the incision size, pore distribution, incision direction and shear loading direction on the shear performance of the GK membrane can be optimized by combining molecular dynamics simulations and machine learning. Besides, according to the latest report [20], the single-atom incised GK membranes could be produced through focused-ion-beam and irradiation-induced techniques. We also plan to further fabricate the single-atom incised GK membrane using the focused-ion-beam technique and test the corresponding shear performance by experimenting.

Overall, this research demonstrated the shear performance of the GK membranes. It may assist the audience in a broad spectrum of fields, including CO_2_ capture and storage, water desalination, wastewater treatment, hazardous waste containment system, chemical processing and bionics to better understand the mechanical properties of the GK membrane for further practical applications.

## Figures and Tables

**Figure 1 membranes-12-00886-f001:**
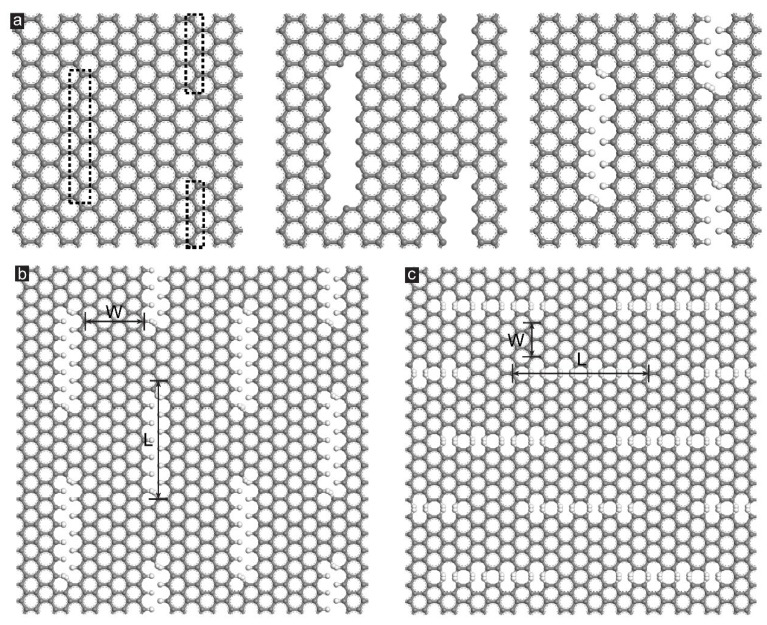
(**a**) Generation of a single-atom incised GK membrane. The carbon atoms in the dotted rectangle were excavated and all dangling bonds were capped using hydrogen atoms. Hydrogenated GK membranes incised in (**b**) zigzag and (**c**) armchair direction.

**Figure 2 membranes-12-00886-f002:**
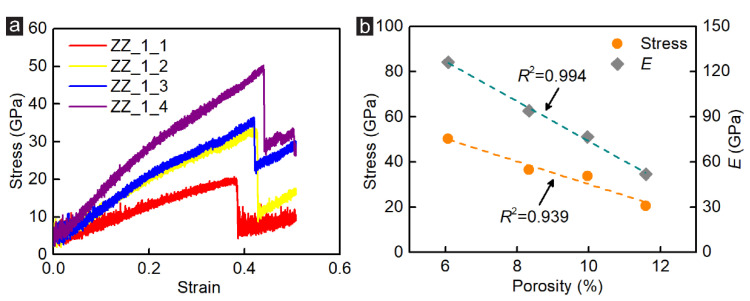
(**a**) The stress-strain curves of the zigzag-incised GK specimens under shear loading simulation; (**b**) The coefficient of stress and shear modulus versus the porosity of GK membrane in Group_1.

**Figure 3 membranes-12-00886-f003:**
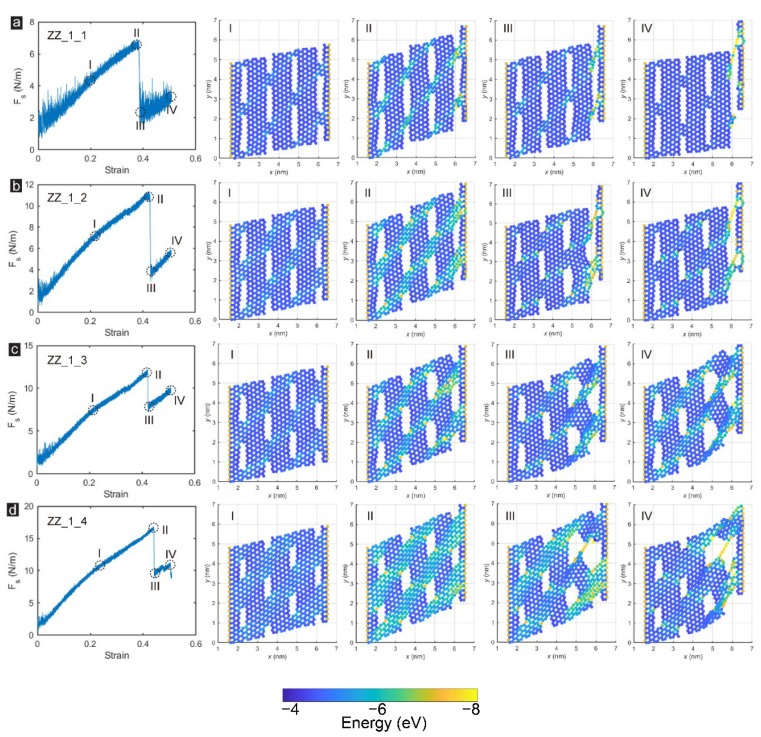
The snapshots of the microstructural damage processes and failure characteristics of zigzag-incised GK membrane: Sample (**a**) ZZ_1_1; (**b**) ZZ_1_2; (**c**) ZZ_1_3 and (**d**) ZZ_1_4.

**Figure 4 membranes-12-00886-f004:**
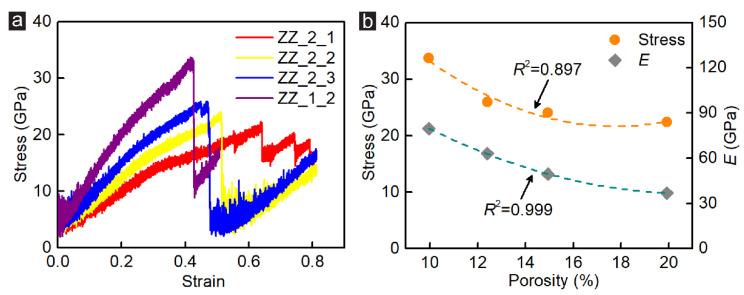
(**a**) The stress-strain curves of the zigzag-incised GK specimens under shear loading simulation; (**b**) The coefficient of stress and shear modulus versus the porosity of GK membrane in Group_2.

**Figure 5 membranes-12-00886-f005:**
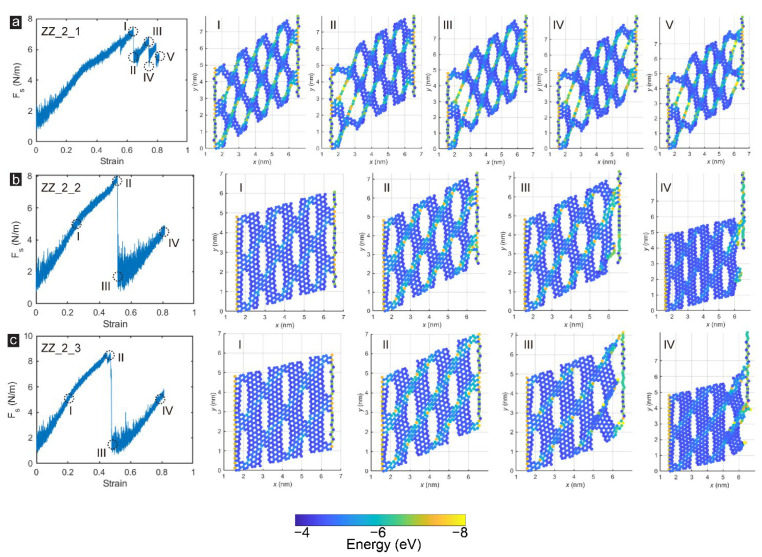
The snapshots of the microstructural damage processes and failure characteristics of zigzag-incised GK membrane: Sample (**a**) ZZ_2_1; (**b**) ZZ_2_2 and (**c**) ZZ_2_3.

**Figure 6 membranes-12-00886-f006:**
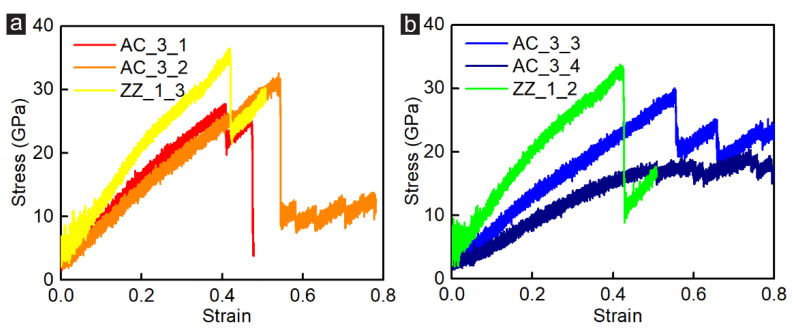
The stress-strain curves of (**a**) Samples AC_3_1, AC_3_2 and ZZ_1_3 and (**b**) Samples AC_3_3, AC_3_4 and ZZ_1_2 under shear loading simulation.

**Figure 7 membranes-12-00886-f007:**
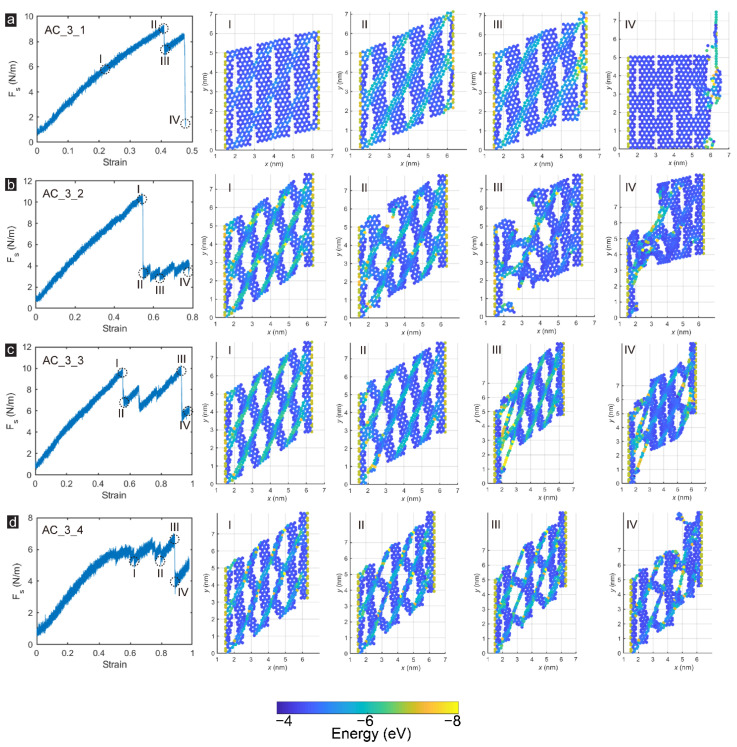
The snapshots of the microstructural damage processes and failure characteristics of armchair-incised GK membrane: Sample (**a**) AC_3_1; (**b**) AC_3_2; (**c**) AC_3_3 and (**d**) AC_3_4.

**Figure 8 membranes-12-00886-f008:**
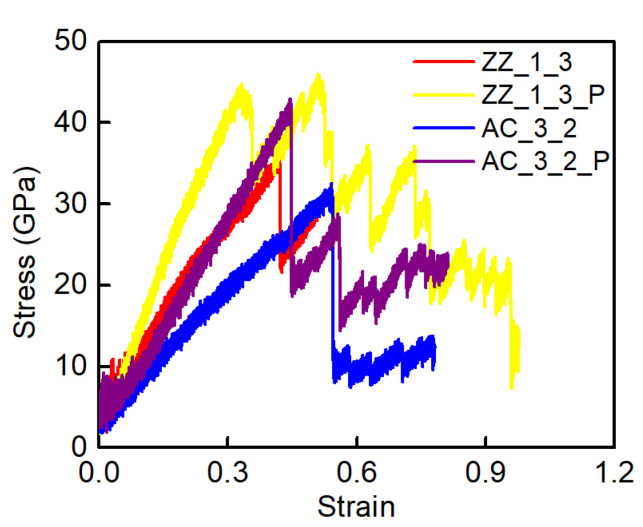
The stress-strain curves of Samples ZZ_1_3 and AC_3_2 under shear loading in both directions parallel and perpendicular to incisions on the GK membranes.

**Figure 9 membranes-12-00886-f009:**
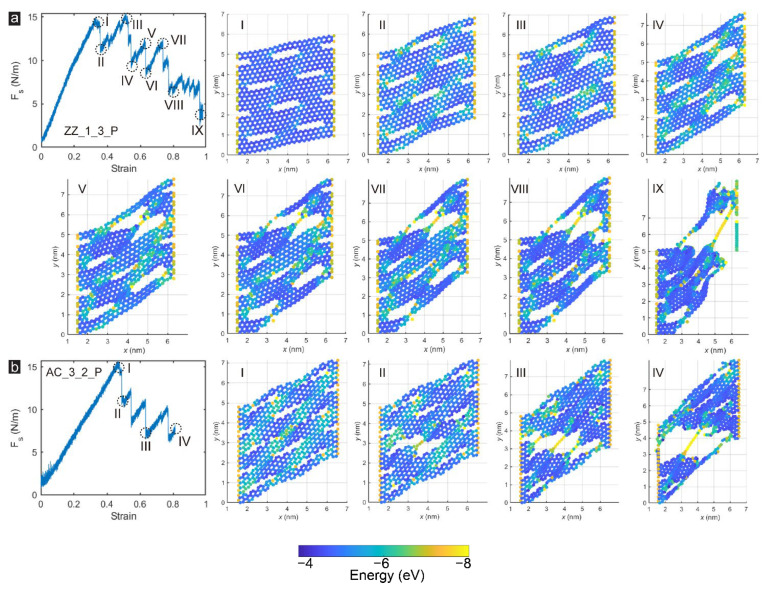
The snapshots of the microstructural damage processes and failure characteristics of the GK membrane: Sample (**a**) ZZ_1_3 and (**b**) AC_3_2 loaded perpendicular to the direction of the incisions.

**Table 1 membranes-12-00886-t001:** Design parameters of the single-atom incised GK membranes.

Sample	L (Å)	D (Å)	Porosity (%)
ZZ_1_1	18.46	8.52	11.59
ZZ_1_2	16.01	8.52	9.96
ZZ_1_3	13.55	8.52	8.33
ZZ_1_4	11.09	8.52	6.07
ZZ_2_1	16.01	2.84	19.92
ZZ_2_2	16.01	4.26	14.94
ZZ_2_3	16.01	7.10	12.4
AC_3_1	19.88	4.92	8.13
AC_3_2	17.04	2.46	8.54
AC_3_3	19.88	2.46	9.76
AC_3_4	21.30	2.46	10.97

**Table 2 membranes-12-00886-t002:** Measured shear properties of the GK membranes based on MD simulation.

Sample	Porosity (%)	Stress (GPa)	Strain	E (GPa)
ZZ_1_1	11.59	20.69	0.38	51.87
ZZ_1_2	9.96	33.73	0.42	79.67
ZZ_1_3	8.33	36.48	0.42	94.00
ZZ_1_4	6.07	50.24	0.44	126.27
ZZ_2_1	19.92	22.36	0.64	36.90
ZZ_2_2	14.94	24.09	0.51	49.40
ZZ_2_3	12.4	25.91	0.47	63.13
AC_3_1	8.13	27.82	0.41	64.85
AC_3_2	8.54	32.51	0.54	51.12
AC_3_3	9.76	30.81	0.93	57.38
AC_3_4	10.97	21.01	0.88	38.00
ZZ_1_3_P	8.33	46.03	0.51	141.15
AC_3_2_P	8.54	42.96	0.45	89.57

## Data Availability

The data presented in this study are available on request from the corresponding author.

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
