# Peer review of "Study on the Shear Behaviour and Fracture Characteristic of Graphene Kirigami Membranes via Molecular Dynamics Simulation"

_membranes, 2022, doi:10.3390/membranes12090886_

Round 1
Reviewer 1 Report
The submission looked at how changing the porosity of graphene kirigami (GK) nanosheets affected their shear strength and elastic modulus by adjusting the incision size, pore distribution, incision direction, carbon atoms, and loading gradient. The shear strength and elastic modulus decreased linearly as porosity increased, according to the authors, when porosity was adjusted by controlling incision size, pore distribution, and incision direction. Similarly, changing the direction of shear loading on GK membranes may improve their shear strength and stiffness. Overall, the manuscript is well-prepared, and the study's findings are intriguing. However, I have a few questions.
1. The phrase “low searchability” appeared several times in the manuscript. What do the authors mean when they say “low searchability”?
2. How can all of the parameters tested in the manuscript be optimized for better shear performance in a GK membrane?
3. How can the modeling results be tested experimentally (in the lab)?
Reviewer 2 Report
In the present work, the author investigated the shear performance and revealed the corresponding structural response and fracture characteristics of the monolayer graphene kirigami membrane. They found that the kirigami structure significantly alters the shear performance of graphene-based sheets. Moreover, they reported that tuning the porosity by controlling the incision size, pore distribution, and incision direction can effectively adjust the shear strength and elastic modulus of GK membranes. I found this paper valuable for the journal readers just after some minor revisions:
1- The abstract section is written too long. The standards of the abstract section should be 150-200 words.
2- Don't use bulk references in the introduction section.
3- The used timestep needs a reference.
4- The authors mentioned that they used the NVT ensemble for all the modeling procedures. Did they use this ensemble for relaxation?? If yes, I believe that they should use the NPT ensemble for relaxation and controlling pressure instead of the NVT because, in the NPT ensemble, the atoms are allowed to move and subsequently eliminate the stress and control the pressure. Otherwise, please mention it in the MD simulation details section.
5- The equilibrium, relaxation, and dynamics simulation time should be mentioned in the MS simulation details section.
6- I believe the initial velocity has essential effects on the results. So, did the authors duplicate the simulation procedure with different initial velocities? If yes, please mention it in the method section; otherwise, you must prove it.
7- The authors are suggested to make the representative input files for MD simulations publicly available to favor the readers to reproduce this work
8- I know that the pristine optimized Tersoff and AIREBO potentials would cause an unphysical strain hardening in the stress values, but I'm not sure about REBO potential. Please check it, and if it causes unusual stress hardening, you need to change the cutoff value and do your MD calculation again.
Reviewer 3 Report
Graphene membrane has attracted great interests in a variety of energy and environmental areas. The proposed research by Gao et al. is interesting and promising for applications. The results and discussions are comprehensive, which can be accepted after minor revisons.
1. For MD simulation details, I wonder whether authors have considered the charges near the dangling bonds and hydrogen atoms. I suspect that the charges probably influence the mechanical properties via the electrostatic interactions.
2. Is the simulation time long enough to get the equilibrium results? How many repeated cases are performed to obtain the equilibrium results.
3. Why porosity is set at the range of 6~12%?
4. In practical membrane applications, multilayer graphenes should have a wider applications than single-layer. Therefore, in addition to the single-layer graphene, authors are suggested to predict or provide some mechanical properties of double-layer or multilayer graphenes.
